# Superior Live Birth Rates, Reducing Sperm DNA Fragmentation (SDF), and Lowering Miscarriage Rates by Using Testicular Sperm Versus Ejaculates in Intracytoplasmic Sperm Injection (ICSI) Cycles from Couples with High SDF: A Systematic Review and Meta-Analysis

**DOI:** 10.3390/biology14020130

**Published:** 2025-01-26

**Authors:** Marina Cano-Extremera, Irene Hervas, Alma Gisbert Iranzo, Mar Falquet Guillem, María Gil Juliá, Ana Navarro-Gomezlechon, Rosa Pacheco-Rendón, Nicolás Garrido Puchalt

**Affiliations:** IVIRMA Global Research Alliance, IVI Foundation, Instituto de Investigación Sanitaria La Fe (IIS La Fe), Avenida Fernando Abril Martorell, 106—Torre A, Planta 1ª, 46026 Valencia, Spain; marina.cano99@gmail.com (M.C.-E.); irene.hervas@ivirma.com (I.H.); almagisbert99@gmail.com (A.G.I.); marfalquetuv@gmail.com (M.F.G.); maria.gil@ivirma.com (M.G.J.); ana.navarro@ivirma.com (A.N.-G.); rosa.pacheco@ivirma.com (R.P.-R.)

**Keywords:** testicular sperm, sperm DNA fragmentation, ejaculated sperm, live birth rate

## Abstract

The use of testicular sperm in non-azoospermic males has emerged in recent years as an attractive option for couples with high sperm DNA fragmentation (SDF) in the ejaculate, repeated ICSI failures, and even poor sperm quality. With this systematic review and meta-analysis, we aim to clarify the findings to date and provide updated information to guide clinical decisions. Our results indicate a clear decrease in the degree of SDF in testicular spermatozoa when compared to ejaculate, and their subsequent use in ICSI cycles leads to a significant increase in the clinical pregnancy rate and a decrease in the miscarriage rate, which is reflected in a significant increase in the rate of live birth at home. In addition, this clinical approach is much more effective in normozoospermic males with high SDF in the ejaculate and with at least one previously failed ICSI cycle. Nonetheless, the findings should be viewed with caution due to the low quality of the studies included and the limited evidence on the safety of this approach for offspring due to chromosome aneuploidies.

## 1. Introduction

Traditionally, semen parameters for assessing male fertility have mainly focused on concentration, motility, and morphology compared with agreed normality thresholds [1]. Thanks to this, we know that low sperm concentration or oligozoospermia affects about 10–15% of men with fertility problems, while more severe conditions such as azoospermia occur in about 1% of the general male population [2]. Since the development of Intracytoplasmic Sperm Injection (ICSI) [3], men with serious infertility factors such as severe oligozoospermia, criptozoospermia, or azoospermia are no longer limited to sperm donation, having the possibility of fathering a biological child using sperm present in ejaculate or by resorting to testicular biopsy.

Nonetheless, conventional semen analysis does not assess intrinsic sperm characteristics that may affect reproductive success at a physiological level. In this regard, the presence of sperm DNA fragmentation (SDF) is one of the male factors proposed to cause impaired outcomes and has been widely studied but with controversial results. This paternal chromatin damage may originate at the testicular or post-testicular level, where damage by external factors is more likely [4]. Some studies showed that semen samples with high SDF influence embryo development and quality, reflected in lower fertilization rates, decreased pregnancy rates, or a high proportion of miscarriages [5,6,7]. However, other studies have not observed such a negative effect on cycle outcomes, finding it especially controversial concerning ICSI cycles. The reason for this controversy may be due to the fact that publications differ in factors such as the patients included or the SFD tests used, indicating the importance of selecting the male population to obtain clinically relevant results [6,8].

It has been noted that infertile males subjected to assisted reproductive treatment (ART) generally have a higher proportion of sperm with elevated levels of fragmentation [9] and that these spermatozoa are equally capable of fertilizing an oocyte as one without genomic damage. In addition, different studies have also reported that high SDF often occurs in conjunction with poor semen parameters [10,11].

To address cases where male DNA integrity is compromised, several strategies have been proposed, focusing on improving sperm quality or selecting sperm with higher chromatin quality. These include, for example, the intervention of varicocele, taking oral antioxidants, or increasing ejaculation frequency [9,12]. All of these can be carried out alone or in combination with a sperm selection technique, such as MACS (Magnetic Activated Cell Sorting), microfluidics, or ICSI, with advanced sperm selection such as PICSI (Physiological Intracytoplasmic Sperm Injection) or IMSI (Intracytoplasmic Morphologically Selected Sperm Injection). Nevertheless, none of these options seems to cause a significant improvement in the results [13].

In addition to the strategies mentioned above, the use of spermatozoa from testicular biopsy versus ejaculate has gained popularity in recent years, as better ICSI results have been reported in males with elevated SDF or with severe oligozoospermia, even in couples with previous ICSI failures using ejaculated sperm [14,15,16,17,18,19,20,21,22,23].

The rationale for using testicular sperm rather than ejaculate sperm is based on the better quality of the first ones due to higher DNA chain integrity. It has been shown, first in an animal model [24] and subsequently in males, that the passage of sperm through the epididymis and male reproductive tract increases SDF due to reactive oxygen species (ROS). Indeed, this has been supported by paired studies where SDF testing has been performed on the same individual in both ejaculated and testicular spermatozoa [14,15,16,17,18,25,26]. Therefore, such post-testicular damage could be avoided by opting for testicular biopsy in men with persistent high SDF after treatment [27].

The first study to test the switch to testicular origin was in 2005, finding a lower SDF in testicular sperm and a significant improvement in clinical outcomes [14]. Subsequently, several studies have also evaluated this approach. In these studies, although similar fertilization rates were found between ICSI cycles with testicular spermatozoa and cycles with ejaculated sperm, a higher pregnancy rate was observed along with a reduction in the miscarriage rate, which was reflected in a significant improvement in the live birth rate [18,20,21,28]. In addition, there are also four published systematic reviews and meta-analyses showing improvements in reproductive outcomes when performing ICSI with testicular biopsy sperm over ICSI using ejaculate sperm [26,27,29,30]. However, these meta-analyses have certain limitations due to the low quality of the studies, the small number of included studies, and their heterogeneity. Furthermore, since the first published meta-analysis, new studies have emerged, showing new information to be taken into account.

For this reason, our goal is to provide the most updated information to date to guide evidence-based clinical decisions regarding the use of testicular sperm instead of ejaculated sperm in ICSI cycles in non-azoospermic men with high SDF and previous ICSI failures or oligozoospermia. In addition, we aim to clearly define the study population in which this strategy significantly improves outcomes.

## 2. Methods

This systematic review and meta-analysis was properly registered in the PROSPERO database ID: CRD42024590921. This registration not only ensures the transparency and accuracy of the research process but also allows other researchers to access information about the methods and objectives of our study.

### 2.1. Search Procedures

A systematic search was performed in Pubmed, Scopus, Web of Science (WoS), Medline, and the clinical trials registry (https://clinicaltrials.gov/ (accessed on 15 November 2024)) according to our study protocol and following the PRIregistration statementSMA (Preferred Reporting Items for Systematic Reviews and Meta-Analyses) guidelines. We compiled all related studies and conference abstracts published until November 2024.

The keywords used for the search were “sperm DNA fragmentation”, “sperm DNA damage”, “ejaculated sperm”, “testicular sperm”, and “intracytoplasmic sperm injection”, always in humans. All study types were considered: observational studies, randomized controlled studies, cross-sectional studies, case–control studies, and journal articles.

### 2.2. Study Selection Criteria

All human clinical studies that were systematic reviews or meta-analyses were not considered. The original studies that were included were those in which: (i) the male had a previous diagnosis of elevated SDF, (ii) ICSI cycles performed with ejaculated sperm (E-ICSI) were compared with ICSI cycles with sperm from testicles (T-ICSI); (iii) the males included in the study had an SDF analysis in ejaculated semen and testicular spermatozoa; and (iv) males with previous ICSI failures with ejaculated semen and or severe infertility (oligozoospermia or ≤15 million spermatozoa/mL). Studies involving males with azoospermia, males without analysis of SDF levels in the ejaculate and/or testis, and using additional sperm selection methods (PICSI, MACS, microfluidics, etc.) were excluded from the selection.

The following information was collected for each study: (i) semen analysis results (normozoospermic versus oligozoospermia), (ii) type of study population (previous failed ICSI cycles versus non-previous ICSI failure), (iii) type of SDF test, and (iv) study design (retrospective, prospective, case–control, randomized controlled study).

### 2.3. Outcomes Measured

The primary outcome was the live birth rate, defined as the delivery of at least one newborn per embryo transfer (ET). The secondary outcomes were SDF levels, fertilization rate (number of two pronuclei zygotes by the total number of microinjected metaphase II oocytes), clinical pregnancy rate (visualization of fetal heartbeat by 7 weeks of gestation by sonography per ET), and miscarriage rate (pregnancy loss after sac visualization per confirmed pregnancy after sac visualization).

### 2.4. Data Analysis

Statistical analysis was performed with Review Manager 5.4.1 software (Cochrane Collaboration, London, UK). A *p*-value < 0.05 was considered significant.

The data were pooled, and for the dichotomous variables, the odds ratio (OR) and 95% confidence interval (CI) were calculated, while for the continuous variables, the standardized mean difference (MD) and 95% CI were used. The I^2^ test was used to quantify the heterogeneity between the studies included, and if the heterogeneity was >50%, the random effects model (REM) test was applied. To reduce heterogeneity in the results, different subgroups were established for each parameter: for SDF, according to the technique used to measure it, and for the rest of the outcomes, the history of previous ICSI failures or semen analysis (normozoospermia or oligozoospermia).

Funnel plots were generated to represent the heterogeneity in the studies, and forest plots were obtained to expose all the included studies for each reproductive outcome assessed.

### 2.5. Risk of Bias Assessment

Funnel plots were constructed using Review Manager 5.4 for qualitative analysis of publication bias based on their symmetry. In addition, because twelve of the thirteen eligible studies did not perform randomization, we used the ROBINS-I (Risk of Bias In Non-randomized Studies—of Interventions) tool that assesses seven domains of bias based on a questionnaire (Appendix A).

## 3. Results

Our search of electronic databases yielded 138 articles, of which 23 were selected for full reading after an initial screening by title and abstract analysis. Of these 23 articles, 10 were excluded for various reasons. Three publications turned out to be conference abstracts, which did not have an associated full paper to analyze their results. Six articles did not meet the inclusion criteria. And finally, one article was excluded as the results were impossible to interpret accurately (Figure 1).

Of the total of thirteen studies included, seven were retrospective, five had a prospective design, and the last was a case series study. The main characteristics of the included studies, such as the design, study population, semen parameters, SDF test used, testicular sperm retrieval method, sperm retrieval complications, and outcome measures, are summarized in Table 1.

### 3.1. Sperm DNA Fragmentation

A total of eight studies compared SDF levels between testicular spermatozoa and ejaculate sperm from the same male, with a total of 530 men tested [14,15,16,17,18,22,25,31].

The overall mean SDF of the included patients was 14.90% ± 12.82 in T-ICSI and 39.62% ± 13.12 in E-ICSI (*p* = 0.002). The meta-analysis indicates that SDF is significantly lower in testis sperm than in ejaculate sperm, with an MD of −25.42 [−31.47; −17.30] and an I^2^ = 90% (*p* = 0.00001) (Figure 2). Given the high heterogeneity in the analysis, the REM test was applied, and in addition, a sub-analysis was performed based on the technique used to measure DNA fragmentation. Five studies used the TUNEL assay to assess SDF in their patients (n = 283). The effect was an MD = −20.20 [−22.43–−17.97], with an I^2^ = 0% (*p* < 0.00001), indicating significantly lower DNA damage in testicular sperm. A similar result was obtained when analyzing the three studies using the SCD test (in a total of 257 males): the MD was −32.31 [−41.19; −23.43], with an I^2^ = 72% (*p* = 0.00001). According to our inclusion criteria, no studies using the SCSA technique could be included.

### 3.2. Clinical Outcomes

The effect of switching sperm origin on the fertilization rate, clinical pregnancy rate, live birth rate, and miscarriage rate was evaluated.

Regarding the fertilization rate, a total of eight studies were included [14,18,19,20,21,22,28,32], in which a total of 6995 oocytes inseminated with both testicular sperm (n = 3880) and ejaculate sperm (n = 3115) were evaluated. The fertilization rate was 59.82% (58.28–61.36) and 69.40% (66.77–71.01) for each group, respectively (*p* < 0.0001). The global fertilization rate showed a similar OR between both groups (OR = 0.83 [0.67–1.03], with an I^2^ index = 74%), making this difference non-significant (*p* = 0.10). To reduce heterogeneity in the analysis, the REM was applied considering male semen quality and previous history of failed ICSI cycles, establishing two subgroups of analysis. In two studies, the male population had oligozoospermia and no previous ICSI failures. In this case, the fertilization rate is favored by the sperm from the ejaculate, OR = 0.61 [0.52–0.71], with an I^2^ = 0% (*p* < 0.00001). In contrast, when the selected population has normozoospermia and at least one previous ICSI failure, there is virtually no difference in the fertilization rate between the two sperm origins, having an OR = 0.94 [0.74–1.20], with I^2^ = 63% (*p* = 0.64) (Figure 3A).

Seven studies were included for the clinical pregnancy rate [14,18,20,21,22,28,32], with 540 ET in total from males with high SDF in the ejaculate: 285 transfers corresponding to the T-ICSI group and 255 transfers from the E-ICSI group. The clinical pregnancy rate was higher in cycles with testicular sperm than in those with ejaculated sperm: 45.61% (39.83–51.39) vs. 30.60% (24.94–36.26) (*p* < 0.0001), with an overall OR of 2.13 [1.35–3.36], I^2^ = 19% (*p* < 0.001). After REM application, the effect of sperm origin was slightly higher on the odds of pregnancy in the normozoospermic group with previous ICSI failures (OR = 2.24 [1.06–4.74], I^2^ = 29%; *p* < 0.03) than in the oligozoospermic group with no previous ICSI failures (OR = 2.13 [1.10–4.11], I^2^ = 39%; *p* = 0.02 (Figure 3B).

Eight studies [14,18,19,20,21,22,28,32] reported miscarriage rates for pregnancies in males with elevated SDF (Figure 3C). A total of 35 miscarriages, including 13 from the testicular sperm group and 22 from the ejaculate sperm group, were included. The results indicate a significant decrease in the probability of miscarriage in the T-ICSI group (9.0% (4.32–13.62) compared to E-ICSI (25.90% (16.58–35.20) (*p* < 0.001), with a general OR = 0.31 [0.14–0.70] and I^2^ = 0% (*p* = 0.004). Although there was no heterogeneity in the outcome, we applied the REM to assess the effect of changing sperm origin in each subgroup. In the case of men with oligozoospermia and no previous ICSI failures, the OR was 0.40 [0.07–2.15], I^2^ = 45%, being non-significant (*p* = 0.29). In the group of normozoospermic males with ICSI failure, the OR was like the overall (OR = 0.31 [0.10–0.98], I^2^ = 0% (*p* = 0.05), although, in this case, the heterogeneity was nil (Figure 3C).

Finally, for the live birth rate, only five studies [18,20,21,28,32] reported such an outcome after 446 ET, including 228 from the T-ICSI group and 218 from the E-ICSI group. In this case, the mean live newborn rate was significantly higher in the T-ICSI group (45.2% (38.73–51.650)) than in the group using ejaculate sperm (25.22% (19.46–31.00)) (*p* < 0.0001), with an OR = 2.40 [1.32–4.36] and heterogeneity I^2^ = 45% (*p* < 0.004). Considering the subgroups established, in men with oligozoospermia and no ICSI failure, the live birth rate was significantly higher when testicular sperm was used, OR = 2.58 [1.53–4.34], I^2^ = 0% (*p* = 0.0004). In contrast, a lesser effect was observed in males with normozoospermia and ICSI failures (OR = 2.29 [0.60–8.64], although, in this case, the difference was not statistically significant (*p* = 0.22) and showed high heterogeneity (I^2^ = 72%) (Figure 3D).

### 3.3. Assessment of Research Quality

As twelve of the thirteen included publications were non-randomized studies, the methodological quality of the trials was assessed using the ROBINS-I tool. As shown in Appendix A, the results of the assessment of all studies indicated that two articles were assessed as having a low risk of bias (high quality), seven articles were assessed as having a moderate risk of bias (moderate quality), and three articles were assessed as having a severe or critical risk of bias (low quality).

Funnel plots were used to analyze the heterogeneity between studies as well as publication bias (Appendix A). Depending on the dispersion of the points in the graph, we can assess the asymmetry in the funnel and know whether there is a relevant publication bias. In general, we did not observe a symmetry in the points in the graph. This could be explained by the presence of publication bias, where studies with non-significant or unfavorable results might not have been published, thus making the number of studies small. However, it may also be due to heterogeneity between studies caused by differences in the design and in the characteristics of the selected populations.

## 4. Discussion

The impact of SDF on male infertility and, therefore, on ART outcomes remains controversial and is one of the most hotly debated topics in reproductive medicine, reporting approximately 1500 results in databases such as Pubmed. Depending on the type of damage caused, breaks are found in a single DNA strand (single) or in both (double), and depending on their extent, the consequences for spermatozoa functionality and the resulting embryo can vary [4]. Furthermore, SDF can occur both in males with normal sperm parameters and in males with significant sperm alterations [33,34].

Some studies show that high levels of SDF decrease clinical pregnancy and miscarriage rates in both IVF and ICSI [8,35], while others indicate that there are no significant differences in these parameters or pregnancy rates when performing ICSI [36,37,38,39]. This controversy can be explained by the great heterogeneity among the published studies, as different study populations are involved, distinct techniques are used to measure DNA damage, and no threshold value has been officially established to define poor reproductive outcomes.

Additionally, the role of female factors, such as oocyte quality and maternal age, further complicates the interpretation of these findings. High-quality oocytes have been shown to mitigate the adverse effects of elevated SDF by facilitating DNA repair during zygote activation. Conversely, advanced maternal age or diminished ovarian reserve may reduce the ability to repair DNA damage, exacerbating its impact on ICSI outcomes [40,41,42]. Therefore, both male and female factors must be considered to better understand the variability in clinical results and optimize treatment strategies.

Despite the current discussion, one of the key clinical challenges is to reduce the levels of damaged paternal chromatin and assess the impact this has on reproductive outcomes in patients with previously detected elevated SDF in the ejaculate. Various clinical strategies have been developed to minimize the use of sperm with fragmented chromatin.

Firstly, all correctable factors causing SDF should be assessed. These include varicocele intervention, treatment of infections, lifestyle factors such as alcohol and tobacco consumption and obesity, environmental toxic exposure, etc. Even taking oral antioxidants may help reduce fragmentation levels and eventually avoid a more invasive option [43].

In addition, there are laboratory techniques that allow the selection and use of sperm with less genomic damage. These include PICSI [44], microfluidics [45], MACS [46], and IMSI [47] [48]. Nevertheless, the existing evidence is limited regarding the ability of these techniques to obtain spermatozoa with improved integrity of the genetic material [49]. Furthermore, randomized controlled clinical trials and meta-analyses comparing live birth rates do not show significant improvements in most cases when advanced sperm selection techniques are employed [50,51].

On the other hand, there is growing evidence in the scientific literature, with the majority of published articles agreeing that the use of testicular sperm could be an effective strategy to select sperm with higher DNA integrity and improve reproductive outcomes in cases of males with sufficient sperm in the ejaculate but with elevated SDF values [14,18,19,21,22,28,52,53], those with poor semen parameters, and even in couples with previous failed ICSI treatments with ejaculated semen [54,55,56,57]. The first group to test such a strategy was Greco and colleagues in 2005 [14], who found a significant improvement in the pregnancy probability of 18 couples with elevated ejaculate SDF.

Spermatozoa acquire a higher state of maturity and fertilization potential during their transit through the male reproductive tract until ejaculation [58]. However, spermatozoa from the testis can also successfully fertilize and give rise to good embryo development if they have completed all maturation stages [59,60,61]. In addition, these spermatozoa are thought to have lower levels of SDF because excessive production of ROS or imbalances in antioxidant mechanisms may cause sperm DNA damage along the genital tract, as empirically demonstrated in animal models and humans [14,15,18,24]. In fact, it has been shown that the percentage of sperm with fragmentation (assessed by TUNEL) increases progressively and significantly from the testis (11.4% ±6.0) to the epididymis (15.8% ± 8.0), vas deferens (20.4% ± 10.0), and ejaculate (32.9% ± 20.0) [25].

The first studies showed that changing the sperm source in couples with recurrent implantation failure, several previous unsuccessful IVF/ICSI cycles [57], and high SDF in the ejaculate [14] significantly improved clinical outcomes in terms of the pregnancy rate with the use of testicular sperm. Since then, several retrospective and prospective studies have been conducted to evaluate the potential of this clinical strategy in males with elevated SDF and or poor semen quality [18,19,20,22,28,52,53,55,56,57]. Actually, improved pregnancy rates with a decreased likelihood of miscarriage, resulting in a higher rate of newborns, have been reported in these studies when testicular sperm are used compared to ejaculate sperm in ICSI cycles.

Nonetheless, it should be noted that all the studies conducted to date, despite the relevant results, are observational and were conducted in non-randomized groups of patients [18,19,22,52], without performing any adjusted analysis for potential confounding variables, or are case-crossover studies [14,17,21,28]. In an attempt to unify the results found, different meta-analyses were carried out. To date, there are four systematic reviews and meta-analyses summarizing the currently available evidence on T-ICSI in men with high SDF in ejaculated sperm [26,27,29,30].

The first meta-analysis published was in 2017 [27], including a total of seven studies in which the superiority of spermatozoa of testicular origin over ejaculate was already demonstrated in selected males with high SDF in the semen ejaculate, with significantly higher odds in the rate of pregnancy and live births, while considerably decreasing the rate of miscarriage, especially in couples with male infertility (oligozoospermia). Since then, more studies have been published, so we included all studies that met the same inclusion criteria to increase the statistical strength of this clinical approach. In total, we included 13 studies, six more than in Esteves’ meta-analysis [27].

Regarding the reproductive results, we evidenced a significant decrease in DNA damage in testicular spermatozoa compared to ejaculated ones after paired measurements in the same patients. Regardless of the different threshold values and measurement methods used (TUNEL and SCD), the results strongly favored intervention (T-ICSI) in non-azoospermic males with elevated SDF levels in the ejaculated semen. It should be considered that the number of studies evaluating testicular and ejaculated SDF in a paired manner is low (we were able to include only eight of the thirteen studies). This could be due to technical difficulties in measuring SDF in biopsy samples or with low sperm counts. Nonetheless, the study by Zhao and colleagues [26] also found a significant decrease in SDF levels in testicular spermatozoa compared to those in the ejaculate.

Following the division made by Esteves et al. [27], we also addressed the clinical outcomes in two different populations depending on whether the males had previous ICSI failures but were normozoospermic or if they had oligozoospermia but no previous ICSI failures. Accordingly, we found that fertilization rates are not significantly different when comparing T-ICSI versus E-ICSI. It has been shown that sperm with high SDF achieve successful fertilization, but the detrimental effects of DNA breakages can be reflected lately in embryo development. This may be due to embryonic cells acquiring karyotype imbalances and generating chaotic mosaicism, which increases the chances of miscarriage [62].

This is subsequently evidenced by a pregnancy rate twice as high in T-ICSI versus E-ICSI cycles, along with a significant decrease in the probability of miscarriage when the sperm origin is changed in both subgroups evaluated, oligozoospermic males and couples with previous ICSI failures. These results support our findings, given that the group of patients treated with T-ICSI (where we assume lower SDF) shows a significantly lower miscarriage rate. Consequently, we found a significantly higher live birth rate in couples who underwent T-ICSI, with up to two times higher odds, although we only found a significant effect in the population with oligozoospermia without previous ICSI failures. Nevertheless, it should be noted that we were able to include a few studies in this analysis because not all of them evaluate this rate, which limits the findings and elevates the heterogeneity.

In contrast, if we compare our results with the meta-analysis by Yu et al. [29], the results differ in the studies included and, therefore, in the populations evaluated. They included studies in which the couples had at least one previous ICSI failure, establishing a subgroup with high SDF in the ejaculate and another subgroup of unselected males, in which the switch to T-ICSI was not based on having a high SDF, and the males could have normal or abnormal semen parameters. In agreement, a higher probability of pregnancy was also found with the use of testicular sperm instead of ejaculated sperm, with this effect being more evident in the population with high SDF. In contrast, this association was not significant in unselected males. Likewise, they found that the live newborn rate increases significantly with the use of testicular spermatozoa in the subgroup with high SDF and not in unselected males. This reinforces the importance of carrying out T-ICSI in a defined population group and not for everyone.

Also, there are two recently published meta-analyses that included a total of eleven and nine articles, respectively [26,30]. They found the same results of the superiority of T-ICSI cycles in males with elevated SDF in ejaculated sperm in terms of clinical pregnancy rates and decreased miscarriage rates. In addition, Zhao’s meta-analysis [26], in contrast to Khoo’s [30], also provides the live birth rate, which is significantly higher using T-ICSI. In these two studies, the analysis was performed considering the technique used to measure SDF in the patients, an important factor to consider as it is a source of heterogeneity between diagnoses and, thus, between studies. This is because there is no consensus on the best strategy for assessing DNA fragmentation or established reference threshold values [63]. However, in the context of human reproduction, it is much more complex to establish cut-off values because reproductive outcomes depend on the fertility of the couple as a whole.

For this reason, the type of SDF test as well as the reference threshold value will always be a limitation and a matter of debate between all studies until there is a consensus. Therefore, Zhao’s and Khoo’s analyses [30,64] provide a correct handling of this source of heterogeneity between study populations, but we also believe that is very important to consider the type of patients to whom this option is applied (in this case, couples with a more or less severe male infertility factor or history of previous cycles) to target treatment to the selected group of males who would benefit the most.

Finally, what we can infer from the results obtained from this and the other existing meta-analyses is that the effect of the intervention on reproductive outcomes is positive; however, the analysis is not without limitations. It should be noted that the number of publications comparing reproductive outcomes of T-ICSI versus E-ICSI in males with high ejaculate fragmentation is still limited, with small sample sizes, and all are cohort studies or case series. Furthermore, the study populations varied between studies, and not all reported the characteristics of the study population, such as maternal and paternal age, duration of infertility, stimulation protocols, number of oocytes retrieved, etc., which are variables that could directly influence reproductive success. We conducted a thorough review of the studies included in this meta-analysis with the aim of performing a meta-regression to adjust for important confounding factors, but we found a great lack of information between the different variables provided by each study, as well as Khoo’s [30].

Therefore, when assessing the quality of evidence of our articles, three were finally categorized with a serious risk of bias, seven with a moderate risk of bias, and only two with a low risk of bias. Furthermore, given our funnel plot results, there might be some bias in the articles included as they are not evenly distributed or symmetrical in the graph, which may limit the robustness of our findings. Nevertheless, we managed to control this discrepancy between studies by reducing the statistical heterogeneity, using subgroup analyses to corroborate that the effect of the intervention was similar between different patient populations (oligozoospermia/non-ICSI failure versus normozoospermia/ICSI failure) and between different SDF measurement techniques.

On the other hand, there is controversial data on the increased rate of aneuploidy in sperm from the testis and in cases of severe male infertility. A correlation between SDF and the presence of chromosomal aneuploidies has also been documented, although another possibility is that errors in sperm meiosis related to chromosomal aneuploidies are the cause of DNA breaks [65,66]. None of the studies included in this meta-analysis explored this question. We only found one publication where they reported a similar euploidy rate between both types of cycles [67] after 940 trophectoderm samples from 362 couples with high SDF in the ejaculate (280 of them underwent ICSI with their own ejaculated sperm and 82 couples underwent ICSI with testicular sperm). The euploidy rate per metaphase II oocyte was 18.7% in E-ICSI and 18.3% in T-ICSI.

Also, in a small retrospective study we conducted [68], we found a comparable rate of aneuploidy between couples who underwent T-ICSI after one or more failed ICSI cycles, but the study population involved were couples with previous ICSI failures with unknown values of male SDF. In these two examples, an equal rate of ploidy was found among embryos from ICSI cycles with testicular spermatozoa as in those from ejaculate; however, the evidence is still insufficient, and this fact should be explored with well-designed studies and larger sample sizes since these results were obtained in small groups of patients. In the meantime, couples opting for this treatment should be properly advised by their clinicians, where it is highly recommended to perform a preimplantation genetic test for aneuploidies (PGT-A) to the generated embryos to increase the safety of the treatment by transferring chromosomally normal embryos. This procedure would increase the total cost of the treatment; therefore, patients should be properly counseled to evaluate the cost–benefit of it to obtain a healthy baby at home.

Finally, we are aware that the clinical decision to use testicular spermatozoa in non-azoospermic patients is still controversial and not without risks, as it is a surgical procedure that may have side effects [69]. However, in cases where chromatin quality remains altered after treatment or where no contributing factor has been identified, testicular biopsy could be considered as the next line of treatment. In this case, the pros and cons should be weighed against each other. The most common side effects are testicular pain, swelling, infection, and hematoma, most of which resolve spontaneously and without sequelae [43]. Specifically, in the studies included in our meta-analyses, the complications associated with the procedure were reported as non-existent or minor and could be assumed, bearing in mind that we demonstrated a significant improvement in reproductive outcomes with a probability of LBR twice as high. This would also save women from the risks of undergoing repeated hormone treatment and punctures with their risk, thus reducing the time until finally having the child at home.

## 5. Conclusions

In conclusion, our study supports the results found to date that sperm from the testis show less DNA damage and their use in non-azoospermic males with elevated SDF in the ejaculate significantly improves their clinical outcomes, increasing their chances of having a pregnancy and newborn at home. In particular, the data show that this clinical approach is much more effective in normozoospermic males with high SDF in the ejaculate and with at least one previously failed ICSI cycle. However, it should be recommended with caution due to the limited evidence available on its effect on possible chromosomal aneuploidy in embryos. Further randomized controlled studies would be needed to support this finding.

## Figures and Tables

**Figure 1 biology-14-00130-f001:**
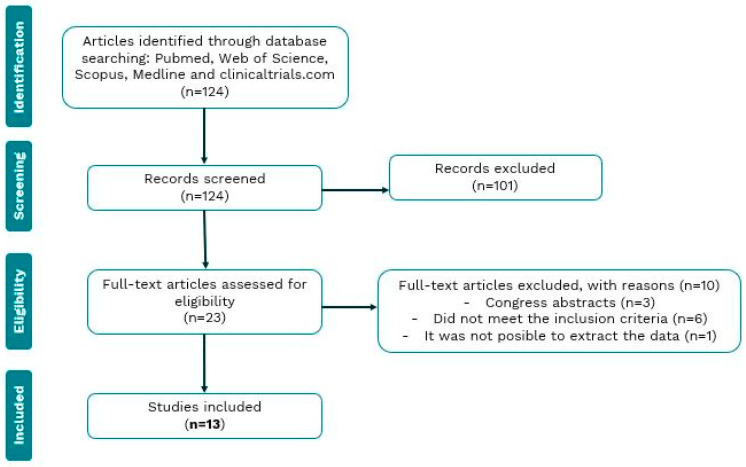
Flow chart of the review process and selection of studies included in the meta-analysis.

**Figure 2 biology-14-00130-f002:**
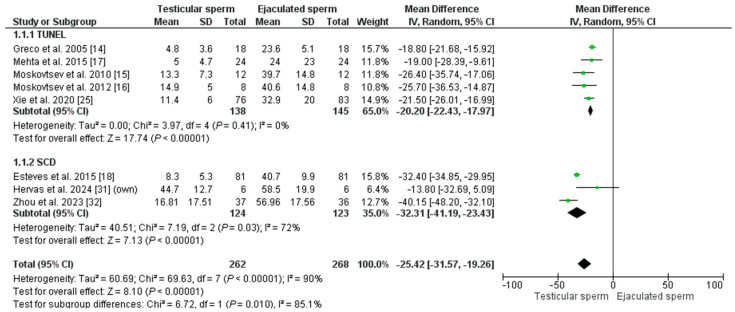
Forest plot showing the mean difference (MD) for sperm DNA fragmentation (SDF) rates between testicular and ejaculated sperm in men with high SDF. Two subgroups are established depending on the technique for measuring SDF: SCD (sperm chromatin dispersion) test and TUNEL test (Terminal deoxynucleotidyl transferase dUTP nick end labeling). CI: confidence interval; IV: inverse variance.

**Figure 3 biology-14-00130-f003:**
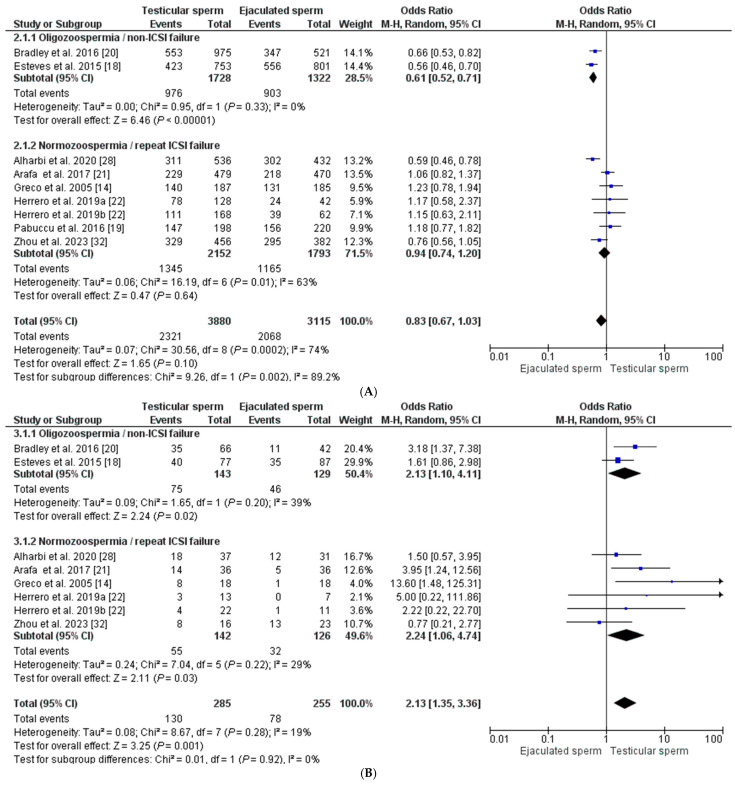
Forest plot showing odds ratios (OR) for each clinical outcome assessed, fertilization rate (**A**), clinical pregnancy rate (**B**), miscarriage rate (**C**), and live birth rate (**D**), between ICSI cycles using testicular sperm and ICSI cycles using ejaculated sperm, in males with high SDF. In addition, two subgroups are established according to the study population (oligozoospermic males with no previous ICSI failures and normozoospermic males with previous ICSI failures). CI: confidence interval; IV: inverse variance. All the overall results obtained are summarized in Appendix A.

**Table 1 biology-14-00130-t001:** Characteristics of the included studies assessing sperm DNA fragmentation (SDF) between ejaculated and testicular sperm and the clinical outcomes of ICSI cycles using testicular sperm (T-ICSI) versus ejaculated sperm (E-ICSI) in males with high ejaculated SDF.

Study	Design	Study Population Criteria: Inclusion (I)/Exclusion (E)	Semen Analysis	SDF Test	Testicular Sperm Retrieval	Sperm Retrieval Complications	Outcome Measures
Greco et al., 2005 [14]	Retrospective cohort study	I: Couples with at least two ICSI failures. Normozoospermic males. N = 18	Concentration (M/mL) (mean ± SD) 26.8 ± 20.8 Motility (%) 36.7 ± 20.1 Mean morphology (%) (media ± SD) 20.9 ± 19.2	TUNEL (>15%)	TESE and TESA	_	FR, CPR
Moskovtsev et al., 2010 [15]	Prospective cohort study	I: Males with persistent high SDF after 3-month antioxidant therapy. E: Varicocelectomy or antibiotic N = 12	_	TUNEL (>30%)	TESE	None	SDF rate between paired testicular sperm and ejaculated sperm
Moskovtsev et al., 2012 [16]	Prospective cohort study	I: Males with persistent high SDF after 3-month antioxidant therapy. T-ICSI: N = 8 E-ICSI: N = 10	Concentration (M/mL) (mean ± SD) 26.7 ± 38.8 Motility (%) 14.1 ± 13.6	TUNEL (>30%)	TESE	None	SDF rate between paired testicular sperm and ejaculated sperm
Mehta et al., 2015 [17]	Case series	I: Oligozoospermic males (<5 million/mL) with at least one IVF or ICSI failure. E: Obstructive azoospermia, non-operative varicocele, testicular trauma, orchiectomy, chemotherapy, or pelvic radiation. N = 24	_	TUNEL (>7%)	micro-TESE	_	SDF rate between paired testicular sperm and ejaculated sperm
Esteves et al., 2015 [18]	Prospective cohort study	I: Infertility > 1 year. Women < 40 years and men < 46 years. Oligozoospermia (<15 million/mL). No physical or endocrine abnormalities. No infections. High SDF in two semen samples after taking antioxidants for at least 3 months. E: Severe oligozoospermia < 5 million/mL and azoospermia. Women with low response to stimulation. T-ICSI: N = 81 E-ICSI: N = 91	Concentration (M/mL) (media ± SD) T-ICSI: 10 ± 3.3 E-ICSI: 9.3 ± 3.9 Motility (%) T-ICSI: 36.6 ± 16.5 E-ICSI: 43.5 ± 11.4 Normal morphology (%) (mean ± SD) T-ICSI: 2.2 ± 2.0 E-ICSI: 2.3 ± 1.8	SCD (>30%)	TESE (n = 29) TESA (n = 52)	They described a 6.2% complication rate (no difference between the TESE vs. TESA group). Pain was the most common complaint (n = 4), while two patients had moderate scrotal swelling.	SDF rate between paired testicular sperm and ejaculated sperm FR, CPR, MR, LBR
Pabuccu et al., 2016 [19]	Retrospective cohort study	I: Normozoospermia. At least two previous ICSI failures. Women aged 18–40 years. Sperm count ≥ 15 million/mL. E: Clinical abnormalities (endocrine profile, physical examination, infections, cancer, cryptorchidism, varicocele, etc.). Women with low response, PGT, oocyte donation, uterine or tubal pathology. Genetic disorders, smokers (>20 cigarettes per day). T-ICSI: N = 31 E-ICSI: N = 40	Motility (range) % T-ICSI: 54.7 (5–200) E-ICSI: 45 (11–87) No. of males with morphology >4% n(%) T-ICSI: 12 (38.7) E-ICSI: 17 (42.5)	TUNEL (>30%)	TESA	None	FR
Bradley et al., 2016 [20]	Retrospective cohort study	I: Oligozoospermia. T-ICSI: N = 148 ciclos E-ICSI: N = 80 ciclos	Mean concentration (range) M/mL E-ICSI: 8.4 (0.6–28.5) T-ICSI: 12.0 (1.6–27.8). Mean progressive motility (range) % E-ICSI: 38 (15–51) T-ICSI: 38 (20–56). Normal morphology (range) % E-ICSI 2 (0–2) T-ICSI 1 (0–2)	SCIT (≥29%)	TESA and TESE	_	FR, CPR, MR, LBR
Arafa et al., 2017 [21]	Prospective cohort study	I: Normozoospermia with previous E-ICSI failure. E: Men with genetic abnormalities, severe oligozoospermia (<5 mill/mL), azoospermia, female factor (low reserve, PCOS, uterine and hormonal pathology). N = 36	Concentration M/mL (mean ± SD) 16.49 ± 20.52 Motility (%) (media ± SD) 19.86 ± 20.65) Abnormal morphology (mean ± SD) 86.25 ± 16.80 SDF (mean ± SD) 56.36 ± 15.3	SCD (≥30%)	TESA	_	FR, CPR, MR, LBR
Herrero et al., 2019 [22]	Retrospective cohort study	I: Normozoospermia with at least two failed ICSI and no previous birth. E: Azoospermia, secondary infertility, donated sperm or oocytes, frozen homologous sperm, non-existent TESE, uterine pathology. T-ICSI: N = 77 E-ICSI: N = 68	Concentration (M/mL) (min–max range) T-ICSI: 37.6 (0.1–271.7) E-ICSI: 51.3 (0.01–246.8) Motility (%) (mean ± SD) T-ICSI:23.5 ± 19.6 E-ICSI: 32.3 ± 24.3 Normal morphology (%) (mean ± SD) T-ICSI: 4.2 ± 6.7 E-ICSI: 4.6 ± 5 NS	TUNEL (>36%) y SCSA (>25%)	TESE	_	CPR, MR, LBR
Alharbi et al., 2020 [28]	Retrospective cohort study	I: Normozoospermia with one or more unsuccessful ICSI cycles. E: Azoospermia, female infertility factor and males with reversible male factor (infections or varicocele). T-ICSI: N = 52 E-ICSI N = 48	Concentration (M/mL) (min–max range) T-ICSI: 22.9 ± 31.6 E-ICSI: 41.2 ± 49.9 Motility (%) (media ± SD) T-ICSI: 42.0 ± 23.9 E-ICSI: 52 ± 22.8 Normal morphology (%) (mean ± SD) T-ICSI: 1.9 ± 1.7 E-ICSI: 1.8 ± 1.6	SCSA (>30%)	TESA	_	CPR, MR, LBR
Xie et al., 2020 [25]	Retrospective cohort study	I: Males with previous ICSI failure and high SDF. E: Males with defects in spermatogenesis with non-obstructive azoospermia or cryptozoospermia. Couples with abnormal BMI, smokers, drug users, or heavy drinkers. T-ICSI: N = 76 E-ICSI: N = 83	-	TUNEL (>15%)	TESE	-	SDF rate between paired testicular sperm and ejaculated sperm
Hervás 2024 [31]	Prospective randomized study	I: Males with severe oligozoospermia (<5 mill/mL) or previous ICSI failure with their partner. Females with adequate ovarian reserve (AMH > 2 ng/mL) or antral follicle count (AFR > 6). E: Abnormal karyotype, microdeletions in the Y chromosome, mutations in the cystic fibrosis gene, varicocele; female age > 38 years, uterine pathology (fibroids, uterine malformations, etc.).	Concentration M/mL (media ± SD) E-ICSI: 5.5 ± 12.02 T-ICSI: 0.09 ± 0.2 Motility (%) (mean ± SD) E-ICSI: 2.67 ± 2.42 T-ICSI: 0.83 ± 0.75	SCD (>30%)	TESE	None	SDF rate between paired testicular sperm and ejaculated sperm
Zhou et al., 2023 [32]	Retrospective cohort study	I: Male patients exhibiting high (25%) SDF. No anomalies in their physical examination or endocrinological profile, and no evidence of genital infection. E: Obstructive azoospermia, unresolved varicocele, testicular trauma, orchiectomy, chemotherapy, or pelvic radiotherapy, any genetic abnormalities.	Concentration (M/mL) (min,max range) T-ICSI: 35.90 (12.0, 88.5) E-ICSI: 13.95 (3.8, 73.7) Motility (%) (min,max range) T-ICSI: 9.60 (1.8, 22.3) E-ICSI: 10.80 (4.3, 23.0) SDF(%) (media ± SD) T-ICSI: 60.58 ± 16.41 E-ICSI: 53.25 ± 18.15	SCD (>25%)	TESA	-	SDF rate between paired testicular sperm and ejaculated sperm FR, CPR, MR, LBR

FR: fertilization rate; CPR: clinical pregnancy rate; MR: miscarriage rate, LBR: live birth rate; ICSI: intracytoplasmic sperm injection; SDF: sperm DNA fragmentation; TESE: testicular sperm extraction; TESA: testicular sperm aspiration; TUNEL: Terminal deoxynucleotidyl transferase dUTP nick end labeling; SCD; sperm chromatin dispersion test; SCSA: Sperm Chromatin Structure Assay.

## Data Availability

Not applicable.

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
