# Peer review of "Superior Live Birth Rates, Reducing Sperm DNA Fragmentation (SDF), and Lowering Miscarriage Rates by Using Testicular Sperm Versus Ejaculates in Intracytoplasmic Sperm Injection (ICSI) Cycles from Couples with High SDF: A Systematic Review and Meta-Analysis"

_biology, 2025, doi:10.3390/biology14020130_

Round 1
Reviewer 1 Report
Comments and Suggestions for Authors
In this study, Cano-Extremera et al. and colleagues reported superior live birth rates, reduced sperm DNA fragmentation (SDF) and lower miscarriage rates by using testicular sperm versus ejaculates in intracytoplasmic sperm injection (ICSI) cycles among couples with high SDF. To evaluate SDF levels between ejaculate and testicular sperm and to compare clinical outcomes of ICSI cycles using testicular spermatozoa (T-ICSI) versus using ejaculate sperm (E-ICSI) in males with previous elevated ejaculate SDF and ICSI failures or severe male infertility, the authors conducted a systemic review using several key terms up to December 2024. In total, thirteen studies met their inclusion criteria. The findings suggest that using testicular sperm in cases of elevated SDF in ejaculated semen with oligozoospermia or previous failed ICSI cycles increases the likelihood of selecting sperm with less DNA damage for ICSI. This improvement is reflected in higher pregnancy rates and reduced miscarriage probabilities, ultimately leading to a significant increase in the likelihood of having a child at home. While the study is well-structured, there are areas that require improvement to enhance the manuscript's overall quality.
Major comments:
1. ICSI outcomes depend on both male and female factors due to the interaction of sperm and oocyte during fertilization. However, the discussion largely focuses on male factors, with limited mention of female contributors. Could the authors expand on or clarify the potential impact of female factors in the discussion section?
Minor comments
1. The language of the manuscript, particularly the abstract, should be polished for clarity and readability.
2. Line 53, ‘other authors’ should be replaced with "other studies" for clarity.
3. Line 143, the ‘I2’ should be changed to ‘I2’. The authors should review the entire manuscript for uniformity.
4. The formatting of Table 1 should be improved for better organization. Additionally, clarify the distinction between "Y" and "y" in the column labeled ‘Testicular sperm retrieval,’ specifically in the studies by Greco et al. (2005) and Bradley et al. (2016)
Comments on the Quality of English LanguageThe language of the manuscript, particularly the abstract, should be polished for clarity and readability.
Author Response
In this study, Cano-Extremera et al. and colleagues reported superior live birth rates, reduced sperm DNA fragmentation (SDF) and lower miscarriage rates by using testicular sperm versus ejaculates in intracytoplasmic sperm injection (ICSI) cycles among couples with high SDF. To evaluate SDF levels between ejaculate and testicular sperm and to compare clinical outcomes of ICSI cycles using testicular spermatozoa (T-ICSI) versus using ejaculate sperm (E-ICSI) in males with previous elevated ejaculate SDF and ICSI failures or severe male infertility, the authors conducted a systemic review using several key terms up to December 2024. In total, thirteen studies met their inclusion criteria. The findings suggest that using testicular sperm in cases of elevated SDF in ejaculated semen with oligozoospermia or previous failed ICSI cycles increases the likelihood of selecting sperm with less DNA damage for ICSI. This improvement is reflected in higher pregnancy rates and reduced miscarriage probabilities, ultimately leading to a significant increase in the likelihood of having a child at home. While the study is well-structured, there are areas that require improvement to enhance the manuscript's overall quality.
First, we would like to thank you for taking the time to read and review our work. We will now reply to each of your questions:
Major comments:
- ICSI outcomes depend on both male and female factors due to the interaction of sperm and oocyte during fertilization. However, the discussion largely focuses on male factors, with limited mention of female contributors. Could the authors expand on or clarify the potential impact of female factors in the discussion section?
Thank you very much for your comment. We have attempted a meta-regression to control for confounding factors, but the included papers lacked sufficient information to be able to do so, including controlling for the female factor. We agree that the female factor is also fundamental in ICSI outcomes and reproductive success. And that especially maternal age or oocyte quality can influence sperm DNA fragmentation effect. We have, therefore, expanded on this in the discussion on lines 303-309.
Minor comments
- The language of the manuscript, particularly the abstract, should be polished for clarity and readability.
We appreciate the comment on the language used. We have improved the manuscript by focusing mainly on the abstract for clarity.
- Line 53, ‘other authors’ should be replaced with "other studies" for clarity.
Thank you very much for the correction, we have applied it
- Line 143, the ‘I2’ should be changed to ‘I2’. The authors should review the entire manuscript for uniformity.
Thank you for your appreciation of this formatting error. We have changed it and made it uniform throughout the manuscript.
- The formatting of Table 1 should be improved for better organization. Additionally, clarify the distinction between "Y" and "y" in the column labeled ‘Testicular sperm retrieval,’ specifically in the studies by Greco et al. (2005) and Bradley et al. (2016).
Thank you very much, we have checked the error and organised the table better for your understanding.
Comments on the Quality of English Language
The language of the manuscript, particularly the abstract, should be polished for clarity and readability.
Reviewer 2 Report
Comments and Suggestions for Authors
1. Due to the low quality of the studies, non-randomized studies,the small number of included studies, and their heterogeneity, the results and conclusion were not consolidated. Therefore, it is better to draw conclusions with caution. At this point, the author handles it well.
2. It is generally believed that conventional semen analysis metrics have a strong correlation with sperm DNA fragmentation (SDF), meaning that high SDF often coincides with poor semen parameters. The value of SDF in assessing male reproductive potential has been widely recognized, but there is no consensus on how significant the results of sperm SDF are in IVF/ICSI. Actually,neither conventional semen analysis nor the SDF results can assess intrinsic sperm characteristics completely that may be affecting reproductive success at a physiological level. The rationale for using testicular sperm rather than ejaculate sperm is still controversial.
3. The topic of this paper has garnered significant attention, with at least four related meta-analyses/systematic reviews published recently. It also presents notable challenges and is highly controversial. Elevated SDF levels often leave both doctors and patients in a difficult position. Shouldn't we first improve spermatogenesis for such patients? Shouldn't we employ sperm selection techniques! Current evidence does not indicate that testicular sperm extraction is necessary for ICSI.
4. The authors should pay more attention to non-invasive methods to deal with the high SDF and supplemented with comprehensive discussion on the topic. Testicular sperm extraction is inherently invasive and destructive, making it difficult to accept from a medical ethics standpoint. After all, how can we explain to patients if ICSI cycles fail? The potential adverse effects following a testicular sperm extraction procedure should also be carefully considered.
Author Response
First, we would like to thank you for taking the time to read and review our work. We will now reply to each of your questions:
- Due to the low quality of the studies, non-randomized studies,the small number of included studies, and their heterogeneity, the results and conclusion were not consolidated. Therefore, it is better to draw conclusions with caution. At this point, the author handles it well.
Thank you for your comment, it is always necessary to highlight the limitations of studies in order to present the results as realistically as possible.
- It is generally believed that conventional semen analysis metrics have a strong correlation with sperm DNA fragmentation (SDF), meaning that high SDF often coincides with poor semen parameters. The value of SDF in assessing male reproductive potential has been widely recognized, but there is no consensus on how significant the results of sperm SDF are in IVF/ICSI. Actually,neither conventional semen analysis nor the SDF results can assess intrinsic sperm characteristics completely that may be affecting reproductive success at a physiological level. The rationale for using testicular sperm rather than ejaculate sperm is still controversial.
Thank you for this reflection, and we totally agree with you that both spermiogram and sperm DNA fragmentation can be used as determinants of a couple's reproductive success. In addition, there is much controversy in the literature about the actual effect of damaged paternal chromatin per se, as there are many factors (such as female or genetic factors) that could be exerting some influence on clinical outcomes.
However, although the use of testicular spermatozoa is equally controversial, mainly due to the fact that they would be obtained by an invasive technique, the results obtained to date indicate a clear improvement in a specific population of infertile patients. However, as we are aware of this controversy, we wanted to make this meta-analysis with the intention of bringing together the results obtained to date and to clarify the size of the effect that this strategy would have for patients with high SDF in the ejaculate, and previous ICSI failures and poor sperm quality. The limited quality of the included studies makes us cautious about the results obtained, but more and more data point to the same point.
- The topic of this paper has garnered significant attention, with at least four related meta-analyses/systematic reviews published recently. It also presents notable challenges and is highly controversial. Elevated SDF levels often leave both doctors and patients in a difficult position. Shouldn't we first improve spermatogenesis for such patients? Shouldn't we employ sperm selection.techniques! Current evidence does not indicate that testicular sperm extraction is necessary for ICSI.
We appreciate your insightful comment, and we agree that improving spermatogenesis is an essential first step for patients with elevated SDF. Strategies such as varicocele intervention, lifestyle modifications, antioxidant supplementation, and addressing underlying medical conditions can help reduce oxidative stress and improve sperm quality. These measures are indeed crucial and should be considered before proceeding to invasive treatments. In parallel, sperm selection techniques such as MACS (magnetic-activated cell sorting), microfluidic sperm sorting, or hyaluronic acid binding have shown promise in selecting sperm with lower levels of DNA damage. However, while these methods may improve the quality of selected sperm, their clinical utility remains limited in cases of persistent high SDF or severe male infertility as we see in the literature.
Our study focuses on a specific subset of patients—those with persistently elevated SDF, and oligozoospermia or previous ICSI failures—where conventional approaches, including improving spermatogenesis or employing sperm selection techniques, may not yield satisfactory outcomes. In these cases, the use of testicular sperm, which often presents lower levels of SDF, has emerged as a viable option. Our meta-analysis explores this approach and demonstrates its potential to improve clinical outcomes.
We acknowledge that the current evidence regarding testicular sperm extraction is not definitive, and we have emphasized this in our manuscript. Further well-designed randomized controlled trials are needed to validate these findings and compare the efficacy of testicular sperm use with advanced sperm selection techniques.
In response to your comment, we have added a section to the discussion in which we analyse all existing strategies for dealing with high fragmentation, their limitations and their importance before moving on to more invasive techniques such as T-ICSI (lines 310-325). We believe this addition provides a more balanced perspective on available strategies and their clinical application.
- The authors should pay more attention to non-invasive methods to deal with the high SDF and supplemented with comprehensive discussion on the topic. Testicular sperm extraction is inherently invasive and destructive, making it difficult to accept from a medical ethics standpoint. After all, how can we explain to patients if ICSI cycles fail? The potential adverse effects following a testicular sperm extraction procedure should also be carefully considered.
We appreciate your thoughtful comment regarding the use of non-invasive methods to manage elevated sperm DNA fragmentation and the ethical considerations surrounding testicular sperm extraction (TESE). We agree that non-invasive approaches should be prioritized whenever possible, and we have revised our discussion to clarify this.
However, in cases where non-invasive methods fail to improve outcomes, such as patients with persistent high SDF, severe oligozoospermia, or repeat ICSI failures with ejaculated sperm, TESE may represent a viable alternative. Our meta-analysis highlights its potential benefits in such scenarios, showing improved clinical outcomes like higher live birth rates, higher pregnancy rates and reduced miscarriage rates when testicular sperm is used.
We acknowledge that TESE is an invasive procedure, and its risks must be carefully weighed against its potential benefits. We also would like to emphasize that TESE should not be the first-line treatment and should be reserved for carefully selected cases where other strategies have proven ineffective. We have tried to make this point clear in the final paragraph of our discussion.
Finally, it should also be noted that ovarian punctures are not without risk. They can have consequences ranging from abdominal pain, haemorrhage, infections, to injury to nearby organs, creation of adhesions, ovarian hyperstimulation syndrome, as well as the emotional and physical stress derived from hormone treatment. While TESE also carries risks (which in the studies reviewed have been found to be minor and manageable) if a single cycle of T-ICSI were successfully completed, we may have avoided the risks of multiple failed ovarian punctures.
Thank you for your valuable feedback, which has allowed us to broaden the scope of our discussion.
Reviewer 3 Report
Comments and Suggestions for Authors
Dear Authors
Congratulations on this excellent work, which is extremely important for anyone who works with ART.
-I did not understand why AA never mention Denny Sakkas and Juan Alvarez.
-I also did not understand why AA did not mention the pioneers in the use of testicular sperm.
-Taking into account the inclusion criteria of having SDF performed on sperm from the ejaculate and the testis, and that cases of azoospermia were excluded, Table 1 appears to include cases with azoospermia and the double execution of SDF is not clear.
Author Response
Dear Authors
Congratulations on this excellent work, which is extremely important for anyone who works with ART.
-I did not understand why AA never mention Denny Sakkas and Juan Alvarez.
-I also did not understand why AA did not mention the pioneers in the use of testicular sperm.
-Taking into account the inclusion criteria of having SDF performed on sperm from the ejaculate and the testis, and that cases of azoospermia were excluded, Table 1 appears to include cases with azoospermia and the double execution of SDF is not clear.
First, we would like to thank you for taking the time to read and review our work. We will now reply to each of your questions:
-I did not understand why AA never mention Denny Sakkas and Juan Alvarez.
Thanks for this suggestion. We have added in the text de reference of their work as it is a great review of the sperm DNA fragmentation topic (reference number 4).
-I also did not understand why AA did not mention the pioneers in the use of testicular sperm.
Thank for this question. We understand this refers to the publication by Greco and colleagues in 2005 in which the use of testicular sperm in couples with elevated SDF in the ejaculate was reported for the first time. We have referred to this work in numerous parts of our paper, being reference number 14. However, we have added a statement in our introduction and discussion section to indicate that it was the pioneering group to suggest this clinical approach in this type of couples (line 94 and 314)E.
-Taking into account the inclusion criteria of having SDF performed on sperm from the ejaculate and the testis, and that cases of azoospermia were excluded, Table 1 appears to include cases with azoospermia and the double execution of SDF is not clear.
Thank you for your observation. Table 1 shows the inclusion and exclusion criteria for each study included in our meta-analysis. We have reviewed table 1 carefully and cases of azoospermia always appear in the excluded population, indicated by the letter E. In no case do they appear in the included populations (I).
Round 2
Reviewer 1 Report
Comments and Suggestions for Authors
The authors have addressed the reviewer's comments comprehensively, and the revised manuscript demonstrates substantial improvement in both clarity and scientific rigor.